# Management of Teeth Affected by Molar Incisor Hypomineralization Using a Resin Infiltration Technique—A Systematic Review

Sylwia Bulanda [1,*] , Danuta Ilczuk-Rypuła [1,*] , Anna Dybek [1] , Daria Pietraszewska [1], Małgorzata Skucha-Nowak [2] and Lidia Postek-Stefańska [1]

1   Department of Pediatric Dentistry, Faculty of Medical Sciences in Zabrze, Medical University of Silesia, 40-055 Katowice, Poland; annadybek20@gmail.com (A.D.); dpietraszewska@sum.edu.pl (D.P.); swrzab@sum.edu.pl (L.P.-S.)
2   Department of Dental Propedeutics, Faculty of Medical Sciences in Zabrze, Medical University of Silesia, 40-055 Katowice, Poland; mskucha-nowak@sum.edu.pl
*   Correspondence: bulanda.sylwia@gmail.com (S.B.); dilczuk-rypula@sum.edu.pl (D.I.-R.)

**Abstract:** In recent years, an increase in children diagnosed with molar incisor hypomineralization (MIH) has been observed. Children with MIH show a high failure rate with conservative treatment. The ICON® system (DMG, Hamburg, Germany), which is an infiltration of decalcified lesions with resin, may strengthen the tooth structure, improve its aesthetics, and cure hypersensitivity. The following article is a systematic review based on the Preferred Reporting Items for Systematic Reviews and Meta-Analyses (PRISMA) protocol. Scientific articles in the PubMed and Google Scholar databases describing the use of the ICON system in the treatment of MIH published in the years 2012–2022 were analyzed. Two independent study authors selected publications that show that the ICON system can be used during the treatment of children with MIH. So far, in the literature, there are no standardized protocols for the dental treatment of patients with hypomineralization of the incisors using the ICON system. Therefore, clinicians rarely use this method of treatment. The ICON system may be successfully used to infiltrate tooth decalcification in children with MIH. However, the depth of infiltration and the achievement of enamel hardness after such therapy are not precisely defined.

**Keywords:** MIH; ICON; molar incisor hypomineralization; dental materials; coatings

## 1. Introduction

### 1.1. MIH

Hypomineralization is one the major symptoms of developmental enamel defects, and it is regarded as an anomaly related to the translucency of tissues. In the case of the presence of molar incisor hypomineralization (MIH), lesions are of concern when appearing in one to four permanent molars, and, additionally, it may occur also in permanent incisors [1]. MIH is diagnosed after the first permanent molars and incisors appear. On the other hand, there is also an analogous syndrome affecting deciduous teeth, which is called hypomineralised secondary primary molars (HSPM) and concerns the second deciduous molars and deciduous canines. Children with HSPM are five times more likely to develop MIH in permanent dentition [2].

The difference in protein composition and lowered enamel mineralization lead to the occurrence of opacity, discoloration, or more severe damage to the hard tissue of teeth [3]. Depending on the stage of advancement of the disease, the correct therapeutical methods of treatment are selected. Non-transparent mottling related to MIH is well circumscribed and opposite to mottling, which is not well circumscribed, with defects which occurr as a result of chronic exposure to a higher gradient of fluoride [4]. MIH causes serious problems for both patients and dentists. Increased enamel porosity weakens the adhesion of the





materials used for fillings. In addition, preventive treatments are more difficult due to the difficulties in achieving a satisfying level of anesthesia. Due to the increased sensitivity to thermal changes, teeth with this disorder are more susceptible to pulpitis. In addition to the above, a very important aspect in the early stages of MIH changes is a lack of aesthetics caused by the occurrence of mottling in the front teeth. People with such changes often feel insecure and do not smile because they are embarrassed about their discolored teeth. Among young people, this aspect may be the cause of depressive disorders and withdrawal from social life. National research estimates the occurrence of MIH in the researched population to be at a level of between 3% and 44% [5]. White spots, which are defined as a demineralization of the surface of the enamel, are a manifestation of the progress of tooth caries, with the possibility to reverse those tooth caries.

This effect is a result of an optical phenomenon because the demineralization process increases the volume of pores, which changes the refractive index (RI) of enamel caused by the presence of water and air [6]. In the progression of this change, a major role is played by factors such as poor oral hygiene, reduced saliva circulation, and the presence or lack of areas of fluoridation. Actions to reduce the defects of enamel, which manifest themselves as white spots, are microabrasion, teeth whitening, resin infiltration, composite restoration, and composite or porcelain veneers [7]. To reduce the loss of hard tooth tissues, the least invasive treatment methods are chosen.

### 1.2. ICON

In recent years, the resin infiltration technique, which can be used without a loss of tooth tissue and without anesthesia, has become a very attractive therapeutic method. It uses a resin of low viscosity, which easily penetrates the etched enamel surface. After infiltration, the material is light-cured.

The refractive index of resin is like that that of enamel, which contributes to the fact that the use of this technique, in many cases, provides a positive end result. The ICON® (DMG, Hamburg, Germany) set comprises of three syringes: the Icon Etch, which is the hydrochloric acid, pyrogenic silica acid, and a surface-active agent; the Icon Dry, which is based on 99% ethanol; and the Icon Infiltrant, which is the infiltrating resin based on methacrylic [8].

The resin infiltrating technique should always be performed in accordance with the manufacturer's recommendations. After the surface of the tooth has been cleaned with the prophylactic paste, the treatment area must be isolated with a rubber dam to achieve dryness and protect the soft tissues. After that, with a use of the microbrush, an Icon Etch gel (15% HCl) is applied. The etch must be left on the surface of the tooth for 2 min. Etching is used to remove superficial mottling and the higher mineralized layer, and to expose the core lesion. The etch is rinsed with water for 30 s and then Icon Dry, which contains ethanol, is applied to dry the surface of the lesion. The Icon Infiltrant, which contains triethylene glycol dimethacrylate (TEGDMA), is applied with the use of a brush and left to penetrate the tissue of the teeth for 5 min. After that, the residue is gently removed and cured with a light. Application of the Icon Infiltrate is repeated 2 to 3 times to reduce the porosity of the enamel. The liquid composite is applied to the lesion and light cured. The last stage is the polishing of the porous surface with rubber discs [9].

The aim of this review was to assess the current approaches in the use of the ICON resin infiltration in the treatment of permanent teeth affected by MIH in children aged 6 years and older and adults. The review is important as it shows alternative, non-invasive treatments that may improve the quality of life of MIH patients. Moreover, the aim of the study was to show a specific clinical approach that includes a modification of the ICON protocol in the treatment of MIH patients.

## 2. Materials and Methods

### 2.1. Protocol Registration

This systematic review has been written based on the protocol created in November of 2021. The scope of the work has been admitted to the international prospective register of systematic reviews, the PROSPERO database, with the identification number CRD42021290606. This systematic review has been conducted in accordance with the guidelines of the Preferred Reporting Items for Systematic Reviews and Meta-Analyses (PRISMA) protocol.

### 2.2. Data Sources and Search Strategy

This systematic review was completed based on the available literature in the PubMed and Google Scholar databases. During searches, the key words used (in accordance with the medical subject headings) were: "molar incisor hypomineralization treatment", "molar incisor mineralization", "ICON", and "resin infiltration". The chosen articles had been written in the English language and published between the years of 2012 and 2022.

### 2.3. Eligibility Criteria

This systematic review was completed based on the PICOS system (population, intervention, comparison, and outcomes) [10]. The strategy for searches and the criteria for inclusion and exclusion from the review were created as set out below.
Search Criteria:

1.    The research group consisted of children and adults with molar incisor hypomineralization, with visible hypomineralization of the front teeth. Research papers describing the early stages of MIH with different etiology were rejected.
2.    The basic criterium for inclusion in the review was the use of the minimally invasive ICON system for the treatment of hypomineralization. Other descriptions of treatment methods were rejected.
3.    The research work papers included randomized clinical trials (RCTs), observational studies, cohort research, retrospectives, and case reports. Only publications in English were included for the review.
4.    Articles describing use of the ICON system have been included in the research.

### 2.4. Study Selection

The study selection was processed in accordance with the Preferred Reporting Items for Systematic Reviews and Meta-Analyses (PRISMA) scheme (Figure 1). The initial selection based on titles and overviews was conducted by two independent authors (S.B. and A.D.). The following information was collected: the type of publication, year of publication, and general description. The extracted data was cross-checked, the disputes were discussed, and agreement was reached. After that, the analysis of full texts was conducted by three authors (S.B., A.D., and D.I.-R.). The authors selected the publications, paying attention to the type of research group, the methodology, and the characteristics of the research related to the application of the ICON system. In cases of non-compliance, the final decision was made by the most experienced member of the research group. The following standard information was extracted from each eligible study: the first author's name, year of publication, type of work, assessed parameters, age of participants, total sample size, and findings. The authors received training before starting the task.

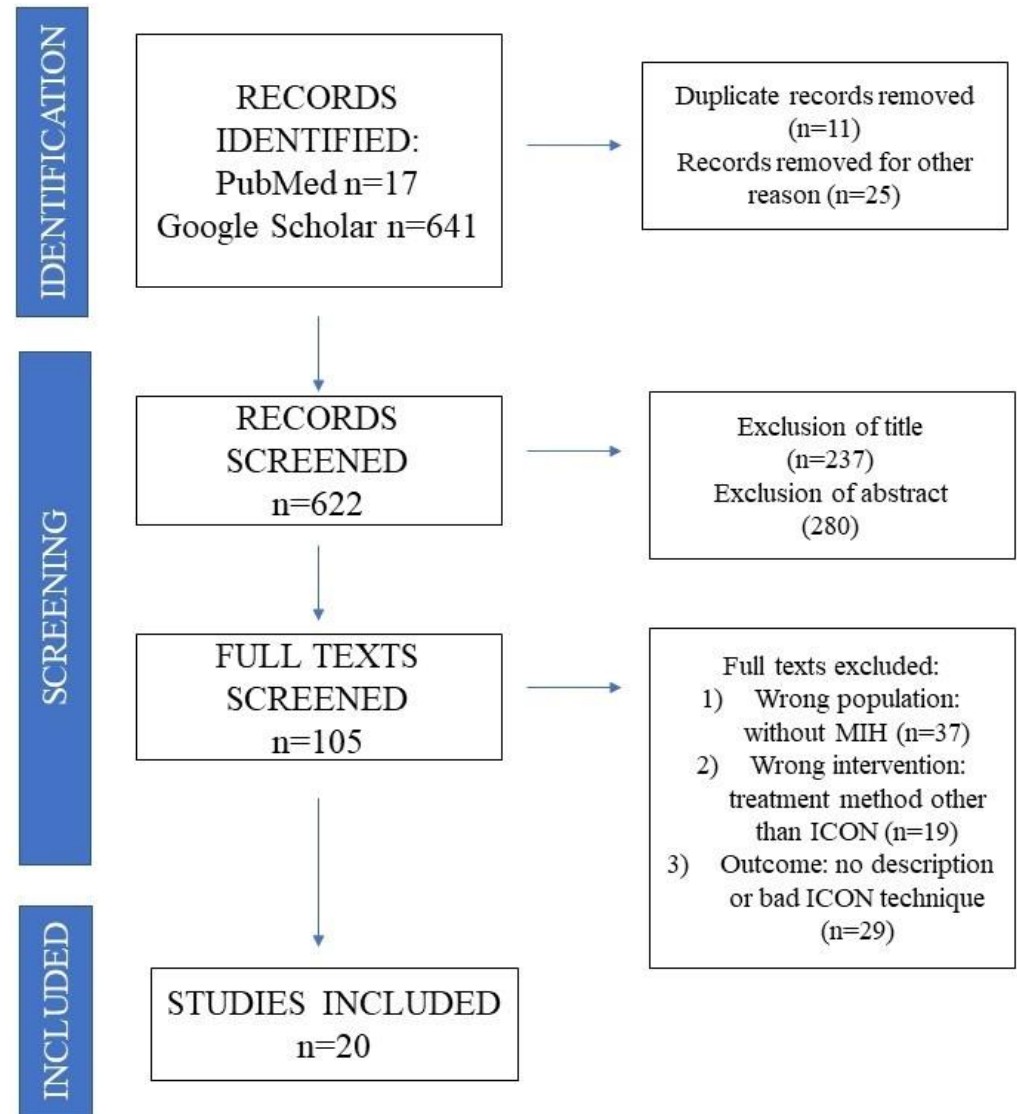

**Figure 1.** PRISMA flowcharts illustrating the inclusion and exclusion criteria.

### 3. Results

In this systematic review, 20 research articles were considered, which describe use of the ICON system in the treatment of molar incisor hypomineralization. The information obtained is presented in Table 1. The following data has been distinguished: the author and year of publication, type of work, information about the research group, assessed parameters, and conclusions from every treatment.

The analysis above showed that use of the ICON system in the treatment of spots on the teeth of MIH patients has gained popularity in recent years. The majority of the analyzed laboratory research, clinical cases, and case reviews (70%) were conducted after 2017. It seems that clinicians have more courage and now more often reach for this treatment method, which can be seen in the five case reviews analyzed, which were completed in recent years.

**Table 1.** The analysis of publications included in the review.

| Publication | Type of Work | Assessed Parameters | Research Group | Findings |
|---|---|---|---|---|
| Kramer, 2018 [11] | In vitro laboratory studies | Assessment of the resin adhesion to hard tissues of a tooth with MIH | 53 freshly extracted human molars and incisors with MIH | ICON can fill porosities of the teeth enamel with MIH but has no impact on the effectiveness of bonding of the enamel with the composite |
| Nogueira, 2021 [12] | Randomized clinical studies | Assessment of the impact of fluoride lacquers or resin infiltration on retaining correct teeth structure affected by MIH | 51 patients aged 6–12 | ICON system had a positive impact on retaining integrity of teeth affected with MIH structure through reduction of risks related to damage caused to enamel |
| Mazur, 2018 [13] | Retrospective research | Assessment of impact of resin infiltration on the aesthetics of teeth with MIH | 76 teeth | Aesthetic effect proved to be highly effective in visual quality and spectrophotometric assessment |
| Marouane, 2021 [14] | Pilot study vivo | Assessment of change of characteristics of resin infiltration over time | Circumscribed MIH lesion with homogenous (14) and non-homogenous appearances (18), patients aged between 10 and 32 years of age | Non-homogeneous lesions require longer application of the infiltrate compared to homogenous lesions |
| Giannetti, 2018 [15] | Monitored clinical research | Assessment of efficiency of superficial infiltration with use of ICON in removal of changes caused by different factors | 17 patients with white cavities of enamel | Cases of MIH should probably be introduced to a more invasive treatment technique |
| Crombie, 2014 [16] | Laboratory research | Research on effectiveness of resin infiltration on lesions in teeth with MIH (with microscope in direction of penetration and with use of SEM microscope) | 21 teeth with lesions similar to characteristics of MIH | Resin infiltration can penetrate changes of enamel with MIH; however, pattern, scope, and change of created hardness are unpredictable |
| Farias, 2022 [17] | Case review | Assessment of effectiveness of ICON system in treatment of lesions with MIH etiology | 20 year old patient with extensive white spots on central incisor | After three sessions of 15% HCI acid erosion, satisfying effect of camouflage of white lesions had been achieved |
| Mabrouk, 2020 [9] | Case review | Assessment of effectiveness of ICON system in treatment of white lesions in enamel with MIH etiology | 29 year old patient with white lesions in enamel on maxillary central incisors | ICON infiltration technique is considered an effective microinvasive procedure in which degree of success is probably achieved due to motivation of patient |
| Diago, 2021 [18] | Cohort research | Research on effectiveness of resin infiltration method in treatment of hypersensitive teeth with MIH | 42 patients aged 8 to 14 with hypersensitivity of minimum on incisor tooth with MIH | Research provides vital and useful initial statistical data and indicates reduction of hypersensitivity in incisor teeth with MIH after only one resin infiltration |
| El-Baz, 2017 [19] | Cohort research | Assessment and comparison of effectiveness of resin infiltration and fluoride lacquer in lesions of MIH etiology (clinical and radiological) | 20 children aged between 9 and 14 | Resin infiltration is significantly better than the fluoride lacquer in masking white spots in children with MIH |
| Bhandari, 2018 [20] | Research in vivo | Assessment of aesthetic treatment results of resin infiltration in incisors with MIH | 22 lesions on incisors | Resin infiltration can mask unaesthetic changes of enamel in teeth with MIH |

**Table 1.** *Cont.*

| Publication | Type of Work | Assessed Parameters | Research Group | Findings |
|---|---|---|---|---|
| Marouane, 2020 [21] | Case review | Impact of use of transillumination during resin infiltration in aesthetic treatment of white spots with MIH etiology | Patient with opacities of enamel within incisors | Use of transillumination can help in achievement of predictable treatment results with use of ICON system |
| Marouane, 2018 [22] | Case review | Assessment of impact of ICON system after teeth whitening of teeth with discoloration with MIH etiology and traumatic etiology | 3 patients: 1 with discoloration of traumatic etiology and 2 with discoloration of MIH etiology | Removal of discoloration, mottling of enamel with use of external whitening constitute a vital stage of initial preparation for resin infiltration |
| Prud'homme, 2017 [23] | Case review | Use of etch-bleach-seal technique in treatment of lesions with MIH characteristics | 3 patients with yellow-brown lesions in incisors | Combining three techniques "microabrasion/etching/whitening/infiltration ICON" might be useful in treatment of yellow-brown mottling frontal in teeth with MIH. |
| Natarajan, 2015 [24] | Laboratory research in vitro | Assessment of pre-treatment preparation with use of HCl, NaOCl, and $H_2O_2$ on depth of penetration of resin infiltrators | 7 molar teeth showing brown mottling, marked without damage to surface of enamel | Pre-treatment preparation helps in deep resin penetration, and MIH part of structure seemed closer to structure of normal enamel |
| Chay, 2013 [25] | Laboratory research in vitro | Research of impact of resin infiltration of ICON on improvement of resin composite adhesion to enamel of teeth with MIH | 152 first permanent molar teeth extracted in children under 18 years old with MIH | Resin infiltration decreases composite bonding to hypomineralized enamel; increased bonding power has been observed in in the initial conditioning of enamel with 5.25% NaOCl, followed by resin infiltration |
| Athayde, 2022 [26] | Randomized controlled clinical trial | Assessment of impact of treatment of restricted opacity in front teeth with use of resin infiltration on aesthetic perceptions in children with MIH (and their parents) | 39 patients aged 8–18 | 15 min resin infiltration can mask opacities in permanent incisors with MIH, bring back aesthetics, and increase self-esteem |
| Hasmun, 2020 [27] | Pilot study | Impact of low invasion aesthetic treatment methods on teeth with MIH on improvement of self-esteem in young patients | 86 children aged 7–16 | After treatment with use of ICON system, self-esteem and social and emotional well-being of patients has increased |
| Singh, 2017 [28] | Cohort research | Assessment of clinical effects of resin infiltration in treatment of hypomineralization of molars and incisors | 12 patients (36 teeth with mild MIH) | Satisfactory aesthetic effect of resin infiltration on treatment of mild hypomineralization of molars and incisors |
| Gandhi, 2012 [29] | Laboratory research | Impact of enamel deproteinization with hypomineralization of molars and incisors (MIH) to sealing resin infiltration | 31 extracted teeth | No difference recorded in treatment with use of prior deproteinization and cases where process has not been performed |

In this systematic review, in vivo and in vitro trials were analyzed. The largest research group during the in vivo trials consisted of 86 patients in the research conducted by Hasum et al. [27]. The largest in vitro group was presented in the research of Chay et al. [25], where 152 extracted teeth from patients with MIH had been infiltrated with resin.

In all the research papers quoted, the results of treating enamel discoloration caused by MIH with the ICON system can be accepted as satisfactory. However, not all cases achieved the full aesthetical aim. The authors suggest that the greater the enamel damage, the lesser the aesthetical outcome. The worst yellow-brown discoloration can be converted into white; however, a 100% aesthetic effect cannot be guaranteed in any case, and hence it seems important to inform patients of the possible failure of the treatment [22,23].

Marouane and Manton described that type 1 changes—homogenous changes—give a positive result for the "ethanol test" (conducted during the application of ICON Dry, wherein the white spot becomes invisible), which shows a better treatment outcome. However, type 2 changes—non-homogeneous changes—evidenced by a negative ethanol test require longer application times because resin infiltration is slower and the aesthetical prospects are restricted [14].

The authors of a few research papers have suggested a modification to the resin infiltration of the ICON system to achieve a better clinical outcome. Prud'homme et al. proposed an etch-bleach-seal technique, which, thanks to the use of microabrasion with 37% phosphoric acid, 5% sodium hypochlorite, and a sealant, can yield a reduction in yellow-brown opacities [23].

On the other hand, Marouane et al. proposed that to completely remove a spot, a whitening procedure must first be performed, followed by resin infiltration of the tissue, which will help in the improvement of the overall look of teeth within a relatively short period of time. On top of that, a greater hardness of the enamel can be achieved, strengthening the weakened histological structure upon which the whitening procedure has been effected [22,30].

In in vitro studies, microscope analysis has shown that the infiltration of unprepared enamel is not great, and it does not reach a depth of 300 μm. The protocols of the initial preparation ensure a more effective infiltration [24]. In the research, where Raman spectroscopy has been used, it has been proven that hypomineralization changes are more likely to look like normal enamel and dentine after deproteinization and demineralization, before resin infiltration [24].

In the research conducted by Nogueira et al. and El-Baz et al., a comparison of the use of the ICON system with fluoride lacquer in the course of MIH treatment had been made. It was proven that resin infiltration with this modification is much better, enabling a minimal invasive treatment during one visit, which is beneficial for adolescent patients [12,19].

## 4. Discussion

MIH syndrome involves lesions that range from small, white enamel defects to yellow-brown opacity in the frontal part of teeth, and it can have a negative psychological impact on patients [26,27,31]. Treatment should be considered, if a patient wants it; however, a consideration of the ratio of the benefits to risks must be evaluated, and the least invasive treatment should be performed [23].

The studied method of treatment is the ICON system of resin infiltration. It is an innovative technology which fills the gap between the prevention and restoration of caries changes to one third of the dentine (D-1) [32]. It has many advantages: it retains the structure of the tooth, its effect is achieved in one visit, a mechanical stabilization of the demineralized enamel occurs, retention of the demineralization occurs, the risk of leakage of fillings and secondary caries is minimized, the aesthetic effect is better, and it is broadly accepted by patients [33–35].

The reason for its better visual effect on the opacities of enamel through resin infiltration is due to the change in the refractive light index. Enamel with hypominerlization has a refractive light index of 1.33; however, a normal light index for enamel is 1.62. When the porosity of the damaged structure is filled with infiltrate, the refractive index rises to 1.52 and the spot becomes invisible [26,36].

Resin infiltration in deciduous teeth is different to that in permanent teeth. The enamel of deciduous teeth is less mineralized, more porous, and aprismatic in comparison to the

enamel of permanent teeth. As a result, the diffusion index appears to be higher for tooth enamel [37]. In the research conducted by Paris et al., deciduous teeth showed a better infiltrative penetration compared to permanent teeth after 1 min of resin application [38].

The rule of resin infiltration relies on the perfusion of porous enamel with resin through the capillary reaction. In this way, the progress of disease is hampered through the disappearance of micropores, which enable acid diffusion paths and melted materials [34].

In MIH syndrome, opacities differ from those of caries etiology or fluoride spots, and hence the outcome of treatment can be different. This change starts from the fusion of enamel with dentine and runs in the direction of the surface of the enamel, from which it achieves a different shape, and it has a narrower surface layer and wider underlayer, creating a sharp angle with the surface of the enamel. The edges of a lesion may be located underneath healthy enamel [39].

Because of this course of action, following the classic ICON application procedure can cause partial etching without full infiltration. To eliminate this issue, the authors suggest a longer etching time, a modification of the etching application, an initial preparation of the structure of the tooth, and a longer infiltration time. Earlier research on caries has shown that the length of application has an impact on the depth and homogeneity of the infiltration. A longer application time has resulted in a deeper penetration of the infiltration [26].

Further, a high level of protein in hypomineralized enamel might be a cause of a more difficult penetration of the infiltrant [14,25]. The use of HCI demineralizes the disordered hydroxyapatites in hypomineralized lesions, leaving a more ordered structure. NaOCl and $H_2O_2$ are helpful in the removal of the excess of protein and peptides, which distort the remineralization of tooth enamel in patients with MIH [24].

During treatment, it is important to consider the thickness of the actual surface layer. The average enamel thickness of front teeth is 300–500 μm in the cervical area, 500–700 μm in the middle part, and 700–1000 μm in the incisal area. Generally, etching is performed 2–4 times, which ensures that the initial erosion achieves a depth of approximately 34 μm, and each etching application increases this depth by 13.28–15.16 μm, with an average depth of erosion of 77 μm [40]. In the research conducted by Gençer et al. [7], Khanna et al. [41], and Bhandhari et al. [20], etching of the enamel was performed for a longer period of time. Only Khann et al. did not show an improvement of the masking effect.

Athayde et al. concluded that after an ICON system treatment, there might be some unwanted side effects, such as after-treatment teeth and gum ache, damage to soft tissues, and a sour aftertaste immediately after the treatment [26]. However, in other publications, none of the authors referred to these problems.

When the structure of a tooth is correctly prepared and the infiltration treatment is conducted correctly, the ICON system can be successfully used up to the D1 depth of damage. In case of D2 and D3 changes, there may be a need for restorative treatment with the use of a composite [42]. However, teeth with MIH are often difficult to correct through reconstruction. Research shows that the bonding force of the composite in teeth with MIH is significantly lower than in teeth with a normal structure [25]. Because of this, a more frequent secondary treatment has been observed for children with MIH [43]. In their research, Chay et al. achieved a better composite bonding to the walls of caries after the initial conditioning of the hypomineralized enamel by using 5.25% NaOCl with further infiltration. Resin infiltration, on its own, reduces the average bonding power of composites [25]. Therefore, increasing the bonding power of a composite with hypomineralized enamel, or giving the weakened structures a greater mechanical resistance through the use of the ICON system, can be very beneficial, even with greater changes, because resin infiltration can prevent acids and bacteria from penetrating through to the deeper tissues located within the structure of a tooth [24].

## 5. Conclusions

The successful identification of an etiological white enamel lesion can provide clinicians with valuable insight into a patient's dentition [33]. The proper control of these

changes remains essential for a correct diagnosis and offering the patient the correct treatment plan. The results concerning the use of the ICON system for the treatment of dental lesions in patients with MIH, as evidenced by the cited studies, are encouraging. In 11 cited studies, the ICON system significantly improved the aesthetics of MIH-affected teeth, and thus also improved the patients' well-being. Three authors pointed out that in the case of deep changes, a good effect is achieved by an earlier preparation of the tooth structure subjected to resin infiltration. All authors cited in this review have suggested that the described treatment carries a risk of failure, and therefore there is still a need for further research into the use of the ICON system in people with MIH.

**Author Contributions:** Conceptualization, D.I.-R. and S.B.; methodology, S.B., A.D. and D.I.-R.; formal analysis, S.B., A.D., D.P. and D.I.-R.; writing—original draft preparation, S.B., A.D., D.P. and D.I.-R.; writing—review and editing, D.I.-R., D.P. and M.S.-N.; supervision, D.I.-R., M.S.-N. and L.P.-S.; project administration, D.I.-R., M.S.-N. and L.P.-S.; funding acquisition, D.I.-R., M.S.-N. and L.P.-S. All authors have read and agreed to the published version of the manuscript.

**Funding:** The research study was performed as part of the authors' employment at the Medical University of Silesia in Katowice, Poland and was funded by grant numbers PCN-1-214/N/1/O and PCN-1-016/N/1/K. The authors would like to thank the Medical University of Silesia in Katowice for the financial support.

**Institutional Review Board Statement:** Not applicable.

**Informed Consent Statement:** Not applicable.

**Data Availability Statement:** Data supporting the findings of the present study can be requested from the authors.

**Conflicts of Interest:** The authors declare no conflict of interest. The funders had no role in the design of the study; in the collection, analyses, or interpretation of data; in the writing of the manuscript; or in the decision to publish the results.

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
