# Peer review of "Management of Teeth Affected by Molar Incisor Hypomineralization Using a Resin Infiltration Technique—A Systematic Review"

_coatings, doi:10.3390/coatings12070964_

Round 1

Reviewer 1 Report

The Authors performed a systematic review to evaluate the management of teeth affected by molar incisor hypomineralization using a resin infiltration technique (ICON system) in children. After eligibility criteria, they found 20 research articles. Analyzing the design, the methodology and the reported results they noted that, although the results of the application of the ICON system for the therapeutic treatment were encouraging, further research is necessary for the assessment of the ICON system as routine treatment.

According to the Authors, this systematic review is performed in accordance with the PRISMA guidelines. However, there are some weaknesses and specific issues which should be addressed.

* Among the most crucial information to include in a systematic review is the question the reviewers plan to investigate. This information (along with the review’s rationale) provides the reader with context and understanding for why the review is being carried out and what the reviewers hope to achieve. According to the Authors (Pag. 2, lines 77-78), “The aim of this review was to assessment of current approaches in the use of ICON resin infiltration in the treatment of teeth affected MIH in children.” However, the search criteria (Pag. 3, line 96) states that “Research group consisted of children and adults”. Why?

* What does “children” mean? The text does not specify the age range of the individuals included in the review. Which teeth were included? Deciduous teeth? Permanent teeth? This information is relevant to be specified in the systematic review.

* Manuscript reports that initial selection of research studies based on titles and overviews were conducted by two independent authors and the analysis of full texts were conducted by three authors. What processes were used to resolve disagreements between data collectors? More details must be given.

* Why a meta-analysis has not been carried out? If it was not possible to conduct a meta-analysis, describe and justify this situation or the synthesis methods or summary approach used.

* Figure 1 and Table 1 show that 20 research articles were included in this systematic review (after eligibility criteria application). However, in Page 4 line 115, the main text states that the systematic review includes 21 articles.

* Lines 176-179, 210, 226. Notice how in UK/US English a decimal point, and not a comma, is placed as separator before the decimal numbers.

* There are few typos/ errors that need attention. For example:

·       Line 108. initial ==> Initial

·       Line 109. […] two independent authors. y (S.B. and A.D.). ?????

·       Line 182. mor porous ==> more porous

·       Line 184. Paris et. Al. ==> Paris et al.

·       Line 203. H2O2 ==> H2O2

·       Etc.

All manuscript must be checked.

Author Response

On behalf of the authors of the work, thank you very much for your valuable comments.
The authors of the study made every effort to improve the text in accordance with the guidelines of the Reviewers.

• The article concerns the use of ICON system in both children and adults diagnosed with MIH. There was a mistake and oversight in the introduction, for which we apologize. The error has been corrected.

The line 30 indicates that MIH applies to permanent teeth. However, due to the ambiguity of the content, the authors supplemented the related information in lines 31-34 and 81.
• As suggested by the Reviewer, information on data selection has been completed in the Study selection section.

This article was constructed as a systematic review. The authors wanted to present various aspects of the problem of treating MIH spots with the ICON system that will be useful for clinicians. The general literature on the subject is limited. It seems that the number of works considered in this review is not statistically significant for the meta-analysis, which is also a suggestion of Reviewer 2.

An error occurred at line 115. Thank you for pointing it out. The authors revised the number of publications included in the review.

  • The spelling of decimal numbers in the indicated lines has been corrected.

  • Typing errors throughout the text have been analyzed and corrected.

On behalf of the authors, I apologize for the omissions and thank you for pointing out the errors and weaknesses of this work. At the same time, we hope that the implemented amendments will improve the quality of the article and make it possible to publish it.

Reviewer 2 Report

The abstract is interesting however all the acronyms should be defined before their first appearance in text.

The introduction has some degree of interest however it will be much better defining actually why is need of this review paper and how this research will improve the state of art and live of patients

What is “CRD42021290606.” ?

“PubMed and Google Scholar” ok but what about industry view/hospitals ?

The quality of image 1 is rather poor as resolution

“21 research articles” it seems rather limited review and work ! not statistically meaningful

The conclusions requires some quantitative details linked to your results and discussions

Author Response

On behalf of the authors of the work, thank you for your valuable comments.
The authors made every effort to improve the text in line with the Reviewers' guidelines.
- All acronyms used in the review have been expanded.
- As suggested in lines 48-50 and 87-90, the clinical benefit of this review has been highlighted.
- The quoted number is the identification number in the PROSPERO system, which was explained in the paper.

- Figure 1 resolution has been improved.

- The authors are aware that the number of reviewed works is not large. However, the general literature on the subject is limited. For this reason, the work is a review and not a meta-analysis.
- Conclusions have been completed in the light of the results obtained.

On behalf of all the authors, I am asking you to re-evaluate the revised work. I hope that all comments have been correctly interpreted and applied in the work.

Round 2

Reviewer 1 Report

The revised version of manuscript is improved, the Authors incorporated the suggestions provided by the reviewers. I thank the Authors for the great effort they have made in responding to the comments of the reviewers. The manuscript can be accepted for publication into Coatings without additional revisions.

Reviewer 2 Report

-